

# Infectious mononucleosis in children and differences in biomarker levels and other features between disease caused by Epstein–Barr virus and other pathogens: a single-center retrospective study in China

Yangcan Ming[1,*], Shengnan Cheng[2,*], Zhixin Chen[1], Wen Su[1], Shuangyan Lu[3], Na Wang[1], Huifu Xu[1], Lizhe Zhang[4], Jing Yu[3] and Jianqiao Tang[4]

[1] Department of Pediatrics, Traditional Chinese and Western Medicine Hospital of Wuhan, Tongji Medical College, Huazhong University of Science and Technology; Wuhan No. 1 Hospital, Wuhan, China

[2] Department of Ophthalmology, Wuhan Hospital of Traditional Chinese and Western Medicine, Tongji Medical College, Huazhong University of Science and Technology; Wuhan No. 1 Hospital, Wuhan, China

[3] Department of Blood Transfusion, Wuhan Hospital of Traditional Chinese and Western Medicine, Tongji Medical College, Huazhong University of Science and Technology; Wuhan No. 1 Hospital, Wuhan, China

[4] Department of Integrated Chinese and Western Medicine, Wuhan Children's Hospital, Tongji Medical College, Huazhong University of Science and Technology, Wuhan, China

[*] These authors contributed equally to this work.

Corresponding authors
Jing Yu, yujings9774@sina.com.cn
Jianqiao Tang, Tjq1996515@126.com

## ABSTRACT

**Background.** Infectious mononucleosis (IM) is a common viral infection that typically presents with fever, pharyngitis and cervical lymphadenopathy. Our aim was to identify the different pathogens causing IM in children admitted to our hospital and to analyze the differences in features of infection with different organisms.

**Methods.** We retrospectively analyzed the data of children aged 0–17 years admitted to Wuhan Children's Hospital during 2013–2022 with IM. We compared symptoms, physical findings, blood counts, and serum biomarkers between patients with IM due to Epstein–Barr virus (EBV) and IM due to other pathogens.

**Results.** Among 1480 enrolled children, 1253 (84.66%) had EBV infection, 806 (54.46%) had *M. pneumoniae* infection, 796 (53.78%) had cytomegalovirus infection, 159 (10.74%) had parvovirus infection, 38 (2.57%) had influenza virus infection, and 25 (1.69%) had adenovirus infection. Receiver operating characteristic curves were used to determine the area under the curve for alanine transaminase (ALT), aspartate transaminase (AST), Alkaline phosphatase (ALP), total bilirubin (TBil), indirect bilirubin (IBil) levels to assess liver damage, and for creatine kinase (CK), CK-MB, and lactate dehydrogenase (LDH) levels to assess myocardial damage. The optimal cutoff values of these biomarkers were then determined. In multivariate analysis, elevated ALT, AST, ALP, TBil, and IBil were independently associated with liver damage, and age <3 years, CK, CK-MB, and LDH with myocardial damage.

**Conclusion.** Evaluation of biomarkers and pathogen detection may help physicians to take preventive actions to avoid serious complications in children with infectious mononucleosis.

## INTRODUCTION

Infectious mononucleosis (IM) is a proliferative infectious disease of the mononuclear macrophage system that typically presents with the clinical triad of fever, pharyngitis, and cervical lymphadenopathy (*Naughton et al., 2021*; *Kien & Ganta, 2020*). Complications include damage to liver and myocardium. The Epstein–Barr virus (EBV), a ubiquitous gamma-herpesvirus that is implicated in a range of hematological malignancies (including Hodgkin lymphoma and Burkitt lymphoma) and with autoimmune diseases such as multiple sclerosis (*Fugl & Andersen, 2019*), is responsible for most cases of IM in children (*Ceraulo & Bytomski, 2019b*). Other known causative organisms include cytomegalovirus (CMV), adenovirus, human herpesvirus-6 (HHV-6), human herpesvirus-7 (HHV-7), and *Toxoplasma gondii* (*Womack & Jimenez, 2015*; *Wang et al., 2010*; *Agut, Bonnafous & Gautheret-Dejean, 2016*). Generally, prognosis is better for IM caused by organisms other than EBV.

Relative to IM caused by EBV (EBV-IM), IM caused by CMV tends to affect patients who are 10-15 years older and to present with milder lymphadenopathy and pharyngitis, though there may be higher likelihood of hepatitis. However, some reports have concluded that EBV-IM and CMV-IM are nearly indistinguishable (*Fulkerson et al., 2021*; *Dunmire, Hogquist & Balfour, 2015*). The populations at risk of EBV infection differ in developed and developing countries. EBV infection is more common in infants and young children in developing countries, but more common in older children and adolescents in developed countries (*Luzuriaga & Sullivan, 2010*).

There are few reviews of IM caused by pathogens other than EBV in infants, children, and adolescents. The aim of this study was to identify the different pathogens responsible for IM in children admitted to our hospital and analyze the potential for liver and myocardial damage in IM caused by different pathogens. We believe that the findings of this study will improve diagnosis and treatment of IM in children and help prevent organ damage.

## METHODS

The data of 1,480 children admitted with IM at Wuhan Children's Hospital, China, during January 2013 to August 2022 were retrospectively analyzed. Children were eligible for inclusion in this study if (1) they had features of the pediatric IM syndrome, namely fever, pharyngitis, cervical lymphadenopathy, hepatomegaly, splenomegaly, and eyelid edema, along with peripheral blood lymphocyte proportion ≥50% and heterotypic lymphocyte proportion ≥10%; (2) active infection as confirmed by detection of the causative organism within 72 h of admission or by detection of EBV serology [negative for anti-Epstein–Barr nuclear antigen (EBNA) IgG antibodies and positive for anti-viral capsid antigen (VCA) IgM antibodies; and (3) the pathogens analyzed were supported by IgM antibodies. The exclusion criteria were (1) severe liver disease (*e.g.*, infantile hepatitis syndrome); (2) concurrent disease requiring hepatobiliary surgery (*e.g.*, biliary atresia); (3) presence of

cardiomyopathy, myocarditis, or other diseases that could affect myocardial enzymes levels; (4) evolution into other diseases, such as hemophagocytic syndrome, chronic active EBV infection, and lymphoma; (5) among the newly diagnosed IM children, the past history or history of present disease has systemic insufficiency of vital organs, so as not to affect the course of disease or test results. The length of hospital stay was also recorded. We accessed data regarding other important factors (*e.g.*, history of present illness and common symptoms, such as fatigue, nausea, vomiting, and the specific sites of the swollen lymph nodes).

## Biomarkers of myocardial and liver damage

Liver damage was diagnosed if the biochemical analyzer (Abbott Laboratories, Abbott Park, IL, USA) showed abnormal values of total bilirubin (TBil), indirect bilirubin (IBil), serum aspartate aminotransferase (AST), alanine aminotransferase (ALT), and/or alkaline phosphatase (ALP); similarly, myocardial damage was diagnosed if there were elevated levels of serum creatine kinase (CK), creatine kinase (CK) isoenzyme MB (CK-MB), and/or lactate dehydrogenase (LDH). Levels of C-reactive protein (CRP) were also detected by the biochemical analyzer (Abbott). Blood cell count analysis was performed using a Mindray BC-6900 hematology analyzer (Mindray Medical International Ltd., Shenzhen, China). The reference values of these markers were set by children's hospital. Differences of various indicators in children of different ages were analyzed.

## Serological tests

Peripheral blood (five mL) was collected and centrifuged at 3,000 rpm for 5 min. Serum levels of anti-EBV (EBV-VCA) IgM and EBNA1-IgG were detected using a chemiluminescence-based IgG antibody detection kit (Yahuilong Biological Technology Co. Ltd., China). IgM/IgG levels above 1 S/CO were considered a positive result. Serum EBV-DNA was detected by real-time polymerase chain reaction using a nucleic acid extraction kit (Daan Orient Gene Biotechnology Company, Shanghai, China) and a genes target detection kit (Daan Orient Gene Biotechnology). CMV-IM was diagnosed if enzyme immunoassay (Boka Biotechnology Company, Shenzhen, China) was positive for CMV-IgM. CMV-IgM levels above 1 S/CO were considered a positive result. Infection by other pathogens (*Mycoplasma pneumoniae*, *Chlamydia pneumoniae*, *Legionella pneumophila*, respiratory syncytial virus, adenovirus, influenza A virus, influenza B virus, and parainfluenza virus) were determined by indirect immunofluorescence (Vircell Company, Spain). Parvovirus IgM was detected by enzyme immunoassay (Boka Biotechnology Company, Shenzhen, China).

This study adhered to the ethical standards of the Helsinki Declaration of 2000 and was approved by the ethical committee of Wuhan Children's Hospital (ethical approval number: 2021R034-E01, the ethical reviewers were Sui Huang and Xuelian He), with waiver of the need for informed consent.

## Statistical analysis

SPSS 25.0 (IBM Corp., Armonk, NY, USA) was used for statistical analysis. The normality of the distribution was compared using Shapiro–Wilk statistics. The Mann–Whitney U-test,
Kruskal–Wallis H-test, one-way ANOVA, independent sample $t$-test, and the chi-square test were used to analyze differences between different groups as necessary. A Bonferroni correction was applied for multiple comparisons. $P < 0.05$ was considered to indicate statistical significance.

Receiver operating characteristic (ROC) analysis was used to identify the optimal cutoff values of the different indicators and to evaluate the specificity and sensitivity of each indicator to predict liver and myocardial damage. Univariate analysis and multivariate Cox regression were used to identify the association of different variables with liver and myocardial damage in patients with IM due to EBV and other pathogens.

## RESULTS

### Characteristics of the study population

All 1,480 children (586 female, 894 male) were diagnosed with IM at the time of admission. The children were in the age range of 0.2 to 16.8 years (mean age, 4.33 years). There were 784 children in the age-group of 0–3 years, 357 in the age-group of 4–6 years, and 339 in the age-group of 7–17 years. At admission, 1,338 children had fever, 1,397 had cervical lymphadenopathy, 1,396 had tonsillitis, 671 had splenomegaly, 611 had evidence of liver injury, and 297 had evidence of myocardial damage. Table 1 shows the characteristics of children with IM due to EBV and IM due to other infections (EBV negative).

Among the 1,480 children, 1,253 (84.66%) had EBV infection (*i.e.,* were EBV-DNA positive or IgM positive), 796 (53.78%) had CMV infection, 806 (54.46%) had *M. pneumoniae* infection, 159 (10.74%) had parvovirus infection, 38 (2.57%) had influenza virus infection, 25 (1.69%) had adenovirus infection, and 1 (0.07%) had chlamydia infection (Fig. 1). Co-infection with EBV and other pathogens was present in 857 (57.91%) children. As Table 1 shows, co-infection was most commonly with CMV (532 patients, 35.95%) and *M pneumoniae* (521 patients, 35.20%). Co-infection with CMV was significantly more common than with *M pneumoniae* in the 0-3-year age-group (345 *vs.* 263, $P < 0.01$) but less common in the 7–17 year age-group (74 *vs.* 138, $P < 0.01$). Clinical manifestations in different age-groups were compared (0–3 years *vs.* 4–6 years *vs.* 7–17 years). Isolated EBV infection (only EBV-DNA positive) was significantly more common in the 7–17-year age-group than in the 0-3-year age-group (81.3% *vs.* 76.4%, $P = 0.037$). CMV infection was significantly more common in the 0–3-year age-group than in the 4–6-year and 7–17-year age-groups (61.8% *vs.* 50.7% and 61.8% *vs.* 42.9%, respectively; $P < 0.001$ for both). However, EBV-IgM positivity and proportion with infection by other pathogens were comparable between the different age-groups. Figure 2 shows the distribution of infection by different pathogens in the study population.

### Comparison of variables between patients with IM due to EBV and IM due to other pathogens

Mean age, atypical lymphocyte count, and the length of hospital stay were significantly higher in EBV-DNA-positive IM patients ($n = 1126$) than in patients with IM due to other pathogens ($n = 83$) ($P < 0.001$ for all). The 1126 IM patients with EBV-DNA positive

**Table 1  Pathogens other than Epstein–Barr virus (EBV) responsible for infectious mononucleosis in children.**

| Pathogen | Primary infection (%) | Uninfected (%) | Co-infection with EBV (%) |
| --- | --- | --- | --- |
| Cytomegalovirus | 264 (17.84%) | 684 (46.21%) | 532 (35.95%) |
| Mycoplasma pneumoniae | 285 (19.26%) | 674 (45.54%) | 521 (35.2%) |
| Parvovirus | 32 (2.16%) | 1321 (89.26%) | 127 (8.58%) |
| Influenza virus | 18 (1.22%) | 1442 (97.43%) | 20 (1.35%) |
| Adenovirus | 9 (0.61%) | 1455 (98.31%) | 16 (1.08%) |

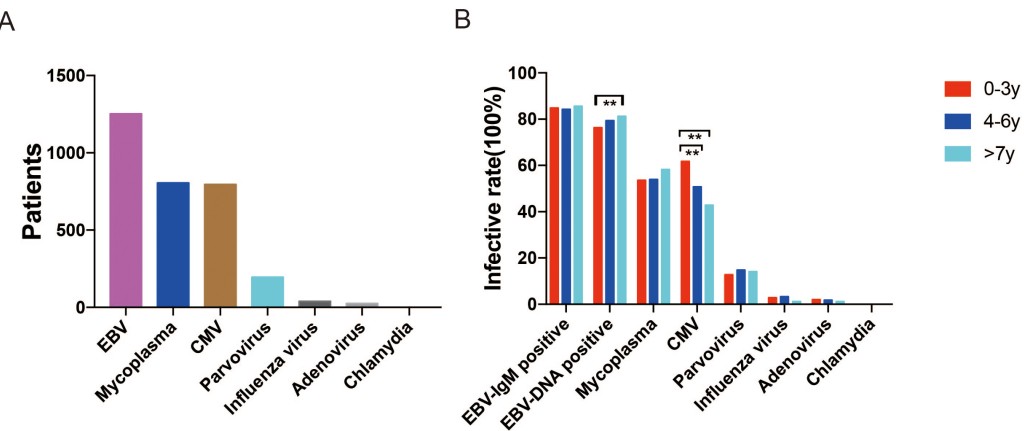

**Figure 1  (A) Pathogens identified in children with infectious mononucleosis during January 1st 2013, to April 26th, 2022, in Wuhan Children's Hospital, China. (B) Proportion of children with different pathogens in the three age-groups.** EBV infection (only EBV-DNA positive) was significantly more common in the 7–17-year age-group than in the 0–3-year age-group (81.25% *vs.* 76.47%, $P = 0.037$). CMV infection was significantly more common in the 0–3-year age-group than in the 4–6-year and 7–17-year age-groups (61.73% *vs.* 50.7% and 61.73% vs.42.77%, respectively, $P < 0.001$).

were detected with positive EBV-IgM. CK-MB level was significantly higher in EBV-DNA-positive IM patients than in patients with IM due to other pathogens ($P < 0.001$). CD3+ and CD8+ T lymphocyte counts were significantly higher in EBV-DNA-positive IM patients than in patients with IM due to other pathogens ($P < 0.001$), but CD4+ T lymphocyte count was significantly lower in EBV-DNA-positive patients ($P = 0.001$). Tonsillitis and cervical lymphadenopathy were more common in EBV-DNA-positive patients than in patients with IM due to other pathogens (tonsillitis: 93.67% *vs.* 87.14%, $P = 0.034$; cervical lymphadenopathy: 93.7% *vs.* 83.9%, $P = 0.004$). Other symptoms were not significantly different between the two groups.

Significant differences were present in white blood cell (WBC) count; lymphocyte count; percentages of CD3+, CD4+, and CD8+ T lymphocyte; and serum TBil level between EBV-IgM- positive patients and those with IM due to other pathogens (Table 2).

## Comparison of different age-groups
Serum CK and CK-MB levels, and WBC and CD4+ T lymphocyte counts, were higher in the 0-3-year age-group than in two older age-groups, while serum ALT, AST, TBil, and IBil

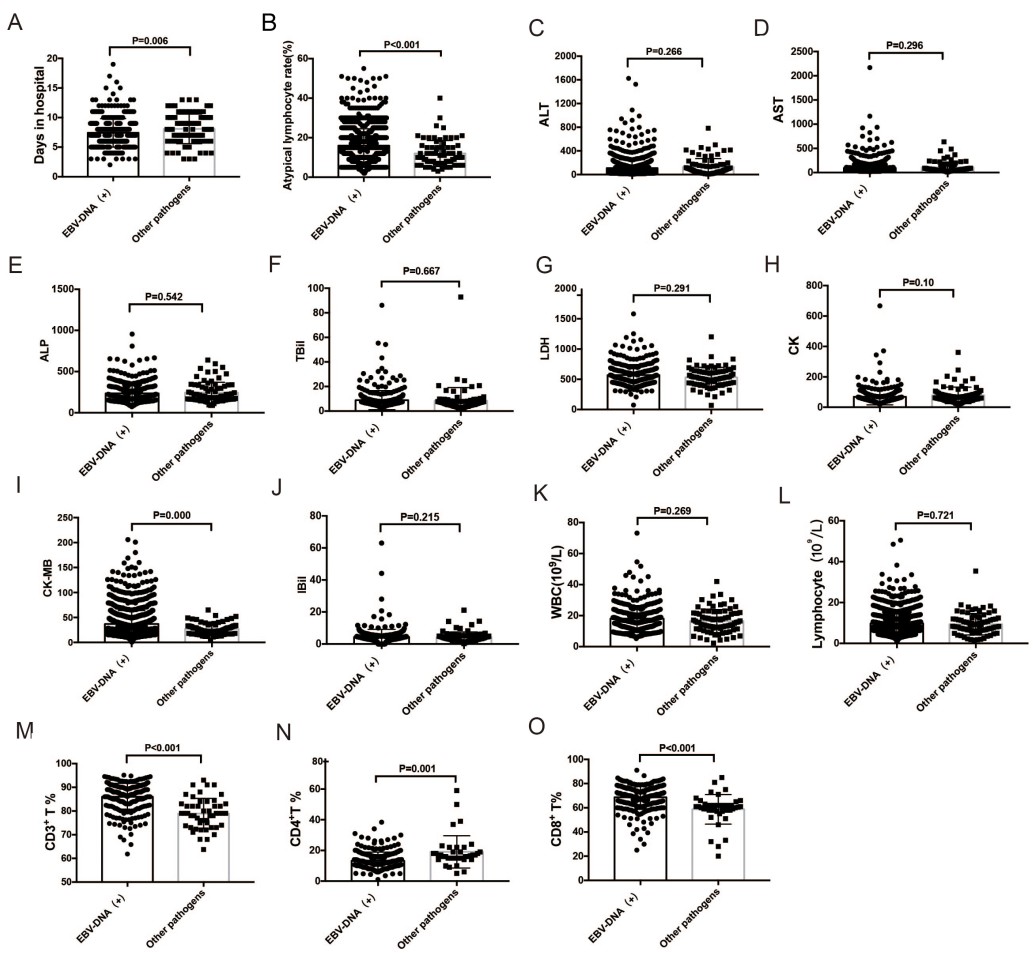

**Figure 2** **Comparison of ALT, AST, ALP, TBil, IBil, LDH, CK, and CK-MB levels; WBC, lymphocyte, and atypical lymphocyte counts; and days in hospital for infectious mononucleosis patients with EBV infection and with other pathogens.** (A–B) Atypical lymphocyte proportion and days in hospital were significantly higher in EBV-DNA-positive patients (all $P < 0.05$). (I, M, N, O) CD3+ and CD8+ T cell proportions and CK-MB level were significantly higher in EBV-DNA-positive patients ($P < 0.001$), while CD4+ T cell proportion was lower. (C–L) There were no significant differences in ALT, AST, ALP, TBil, IBil, LDH, and CK levels, and WBC and lymphocyte counts between EBV-DNA-positive patients and those with IM due to other pathogens. ALT, alanine transaminase; AST, aspartate transaminase; ALP, alkaline phosphatase; TBil, total bilirubin ; IBil, indirect bilirubin; LDH, lactate dehydrogenase; CK, creatine kinase; CK-MB creatine kinase-MB; WBC, white blood cell; EBV, Epstein–Barr virus.

levels, and atypical lymphocyte and CD3+ and CD8+ T lymphocyte counts, were lower (Table 3).

## Cutoff values of biomarkers to assess the liver and myocardial damage in IM patients

Figure 3A shows the ROC curves for ALT, AST, ALP, TBil, and IBil (used to evaluate liver damage), while Fig. 3B shows the ROC curves for CK, CK-MB, AST, and LDH (used to evaluate myocardial damage). The AUC for these variables were as follows: ALT ($0.776 \pm 0.057$) >AST ($0.766 \pm 0.037$) >ALP ($0.746 \pm 0.055$) >TBil ($0.716 \pm 0.039$) >IBil

**Table 2  Clinical manifestations of infectious mononucleosis in children with Epstein–Barr virus (EBV) or other pathogens.**

| Clinical manifestations | All patients $N = 1480$ | EBV-IM $n = 1253$ | Other pathogens-IM $n = 227$ | P value |
|---|---|---|---|---|
| Age, years, mean ± SD | 4.33 ± 2.63 | 4.37 ± 2.67 | 4.16 ± 2.49 | 0.367 |
| Fever | 90.47% | 90.58% | 89.87% | 0.739 |
| Days in hospital, mean ± SD | 7.59 ± 2.45 | 7.62 ± 2.43 | 6.88 ± 2.76 | 0.139 |
| Tonsillar exudate | 94.39% | 94.25% | 95.15% | 0.586 |
| Cervical lymphadenopathy | 94.46% | 94.25% | 95.59% | 0.415 |
| Hepatosplenomegaly | 78.24% | 77.97% | 79.74% | 0.554 |
| Myocardial damage | 20.07% | 19.55% | 22.91% | 0.246 |
| White blood cells ($10^9$/L) | 17.67 ± 8.07 | 17.87 ± 8.09 | 13.22 ± 6.32 | 0.010 |
| Lymphocytes ($10^9$/L) | 9.78 ± 5.22 | 9.88 ± 5.24 | 9.26 ± 5.13 | 0.043 |
| Atypical lymphocytes (%) | 13.45 ± 9.24 | 13.71 ± 9.37 | 13.84 ± 8.49 | 0.536 |
| CRP (mg/L) | 12.34 ± 5.42 | 13.48 ± 4.86 | 13.79 ± 5.02 | 0.567 |
| ALT (U/L) | 109.87 ± 3.66 | 109.01 ± 3.93 | 114.66 ± 9.99 | 0.658 |
| AST (U/L) | 96.05 ± 3.02 | 94.27 ± 3.22 | 105.85 ± 8.52 | 0.503 |
| TBil (μmol/L) | 8.62 ± 0.33 | 8.69 ± 0.35 | 7.01 ± 1.18 | 0.016 |
| IBil (μmol/L) | 4.70 ± 0.18 | 4.73 ± 0.19 | 4.03 ± 0.72 | 0.075 |
| ALP (U/L) | 236.02 ± 5.21 | 235.49 ± 5.24 | 247.43 ± 33.21 | 0.427 |
| LDH (U/L) | 562.14 ± 7.22 | 565.62 ± 7.31 | 487.83 ± 38.59 | 0.026 |
| CK (U/L) | 71.32 ± 2.40 | 71.09 ± 2.49 | 76.04 ± 7.71 | 0.670 |
| CK-MB (U/L) | 36.43 ± 0.68 | 36.13 ± 0.74 | 38.07 ± 1.72 | 0.300 |
| CD3+ T lymphocytes (%) | 85.51 ± 6.01 | 85.79 ± 5.64 | 78.54 ± 9.84 | 0.021 |
| CD8+ T lymphocytes (%) | 67.68 ± 10.79 | 68.27 ± 9.88 | 47.54 ± 19.48 | 0.013 |
| CD4+ T lymphocytes (%) | 14.00 ± 6.82 | 13.50 ± 5.68 | 30.80 ± 16.38 | <0.001 |

Notes.

SD, standard deviation; CRP, C-reactive protein; ALT, alanine transaminase; AST, aspartate transaminase; TBil, total bilirubin; IBil, indirect bilirubin; ALP, alkaline phosphatase; LDH, lactate dehydrogenase; CK, creatine kinase; CK-MB, creatine kinase-MB.

(0.709 ± 0.039) (Fig. 3A); LDH (0.71 ± 0.043) >CK (0.68 ± 0.04) >CK-MB (0.579 ± 0.022) >AST (0.575 ± 0.022) (Fig. 3B). The optimal cutoff values for these markers to predict liver or myocardial damage in IM were as follows: ALT >55.5 U/L, AST >56 U/L, ALP >206.5 U/L, TBil >8.05 μmol/L, IBil >3.65 μmol/L; CK >70.5 U/L, CK-MB >33.5 U/L, AST >56.5 U/L, and LDH >609 U/L.

## Association between different variables and IM-induced liver or myocardial damage

Univariate and multivariate Cox regression analysis were used to identify the association between elevated biochemical parameters and liver or myocardial damage in patients with EBV-IM and IM due to other pathogens (Tables 4 and 5). Figure 4 shows the forest plots. The factors independently associated with liver damage in patients with IM were ALT >55.5 U/L (HR = 4.31, 95% CI [3.23–5.76], $P < 0.001$); AST >56 U/L (HR = 5.26, 95% CI [3.95–7.01], $P < 0.001$); ALP >206.5 U/L (HR = 2.31, 95% CI [1.77–3.02], $P < 0.001$); TBil >8.05 μmol/L (HR = 1.53, 95% CI [1.17–2.02], $P = 0.001$); and IBil >3.65 μmol/L (HR = 1.32, 95% CI [1.0–1.74], $P = 0.045$). Meanwhile, the factors independently associated with

**Table 3  Clinical manifestations of infectious mononucleosis in children of different ages.**

| Clinical manifestations | Total patients N = 1480 | 0–3 years n = 784 | 4–6 years n = 357 | 7–17 years n = 339 | P value |
|---|---|---|---|---|---|
| Fever | 90.47% | 90.31% | 91.29% | 89.97% | 0.999[a], 0.999[*], 0.999[b] |
| Days in hospital | 7.59 ± 2.45 | 7.43 ± 2.39 | 7.75 ± 2.32 | 7.82 ± 2.73 | 0.724[a], 0.999[*], 0.999[b] |
| Tonsillar exudate | 94.39% | 94.13% | 94.38% | 94.99% | 0.999[a], 0.999[*], 0.999[b] |
| Cervical lymphadenopathy | 94.46% | 93.62% | 95.51% | 95.28% | 0.594[a], 0.796[*], 0.999[b] |
| Hepatosplenomegaly | 78.24% | 79.46% | 78.99% | 74.63% | 0.999[a], 0.215[*], 0.491[b] |
| Myocardial damage | 20.07% | 23.85% | 16.53% | 15.05% | 0.013[a], 0.002[*], 1.000[b] |
| WBC count ($10^9$/L) | 17.67 ± 8.07 | 19.14 ± 9.03 | 16.8 ± 7.10 | 14.80 ± 5.07 | 0.044[a], <0.001[*], 0.132[b] |
| Lymphocytes count ($10^9$/L) | 9.78 ± 5.22 | 10.26 ± 5.56 | 9.29 ± 4.85 | 9.19 ± 4.68 | 0.029[a], 0.040[*], 1.000[b] |
| Atypical lymphocytes (%) | 13.73 ± 9.24 | 13.16 ± 9.07 | 13.89 ± 9.20 | 14.89 ± 9.60 | 0.494[a], 0.011[*], 0.565[b] |
| CRP (mg/L) | 13.01 ± 4.56 | 13.45 ± 5.32 | 13.65 ± 4.68 | 12.97 ± 5.23 | 0.432[a], 0.564[*], 0.673[b] |
| ALT (U/L) | 109.87 ± 3.66 | 103.26 ± 4.71 | 107.55 ± 8.26 | 127.61 ± 7.78 | 1.000[a], 0.008[*], 0.006[b] |
| AST (U/L) | 96.05 ± 3.02 | 90.72 ± 4.03 | 96.70 ± 6.57 | 107.70 ± 6.24 | 0.733[a], 0.011[*], 0.019[b] |
| TBil (μmol/L) | 8.62 ± 0.33 | 7.39 ± 0.27 | 7.83 ± 0.36 | 12.74 ± 1.30 | 0.127[a], <0.001[*], <0.001[b] |
| IBil (μmol/L) | 4.70 ± 0.18 | 4.34 ± 0.25 | 4.26 ± 0.20 | 6.14 ± 0.51 | 1.000[a], <0.001[*], <0.001[b] |
| ALP (U/L) | 236.02 ± 5.21 | 228.69 ± 6.53 | 240.85 ± 12.23 | 250.17 ± 11.27 | 0.999[a], 0.205[*], 0.999[b] |
| LDH (U/L) | 562.14 ± 7.22 | 564.71 ± 8.85 | 571.14 ± 17.65 | 544.89 ± 15.25 | 0.999[a], 0.606[*], 0.999[b] |
| CK (U/L) | 71.32 ± 2.40 | 74.78 ± 3.17 | 68.92 ± 3.92 | 65.10 ± 6.46 | 0.302[a], 0.001[*], 0.247[b] |
| CK-MB (U/L) | 36.43 ± 0.68 | 38.11 ± 0.92 | 34.63 ± 1.36 | 34.41 ± 1.46 | 0.002[a], <0.001[*], 1.000[b] |
| CD3+ T lymphocytes (%) | 85.51 ± 6.01 | 83.46 ± 6.27 | 87.21 ± 4.74 | 88.37 ± 4.82 | <0.001[a], <0.001[*], 0.241[b] |
| CD8+T Lymphocytes (%) | 67.68 ± 10.79 | 64.79 ± 10.92 | 70.22 ± 9.13 | 71.52 ± 10.38 | <0.001[a], <0.001[*], 0.874[b] |
| CD4+ T lymphocytes (%) | 14.00 ± 6.82 | 15.16 ± 6.85 | 12.89 ± 5.19 | 12.55 ± 7.85 | 0.025[a], <0.001[*], 0.731[b] |

**Notes.**
[a] comparison of 0–3-years age-group vs. 4–6-years age-group.
[*] comparison of 0–3-years age-group vs. 7–17-year age-group.
[b] comparison of 4–6-year age-group vs. 7–17-years age-group.
WBC, White blood cells; CRP, C-reactive protein; ALT, alanine transaminase; AST, aspartate transaminase; TBil, total bilirubin; IBil, indirect bilirubin; ALP, alkaline phosphatase; LDH, lactate dehydrogenase; CK, creatine kinase; CK-MB, creatine kinase-MB.

myocardial damage were age below 3 years (HR = 1.91, 95% CI [1.26–2.88], $P$ = 0.003); CK >70.5 U/L (HR = 2.83, 95% CI [1.89–4.24], $P$ < 0.001); CK-MB >33.5 U/L (HR = 3.41, 95% CI [2.26–5.12], $P$ < 0.001); and LDH >609 U/L (HR = 2.09, 95% CI [1.41–3.11], $P$ < 0.001).

# DISCUSSION

Infectious mononucleosis, first identified in 1920, is a common disease in children, adolescents, and adults (*Lennon, Crotty & Fenton, 2015*; *Wu et al., 2020*). It is most commonly associated with EBV infection, but about 10% of cases are due to other organisms (*Wu et al., 2020*; *Ishii et al., 2019*; *Medović et al., 2016*). Laboratory findings include lymphocytosis, increased atypical lymphocyte counts, and heterotropic or EBV-specific antibody responses. Serological evidence of EBV-specific antibodies is important evidence for diagnosis of EBV-IM. The most valuable serological finding is anti-virus-capsid antigens (VCA)-IgM antibodies against EBV, which is considered confirmatory for acute primary EBV infection (*Karsten et al., 2022*; *Ceraulo & Bytomski, 2019a*). In our study, all

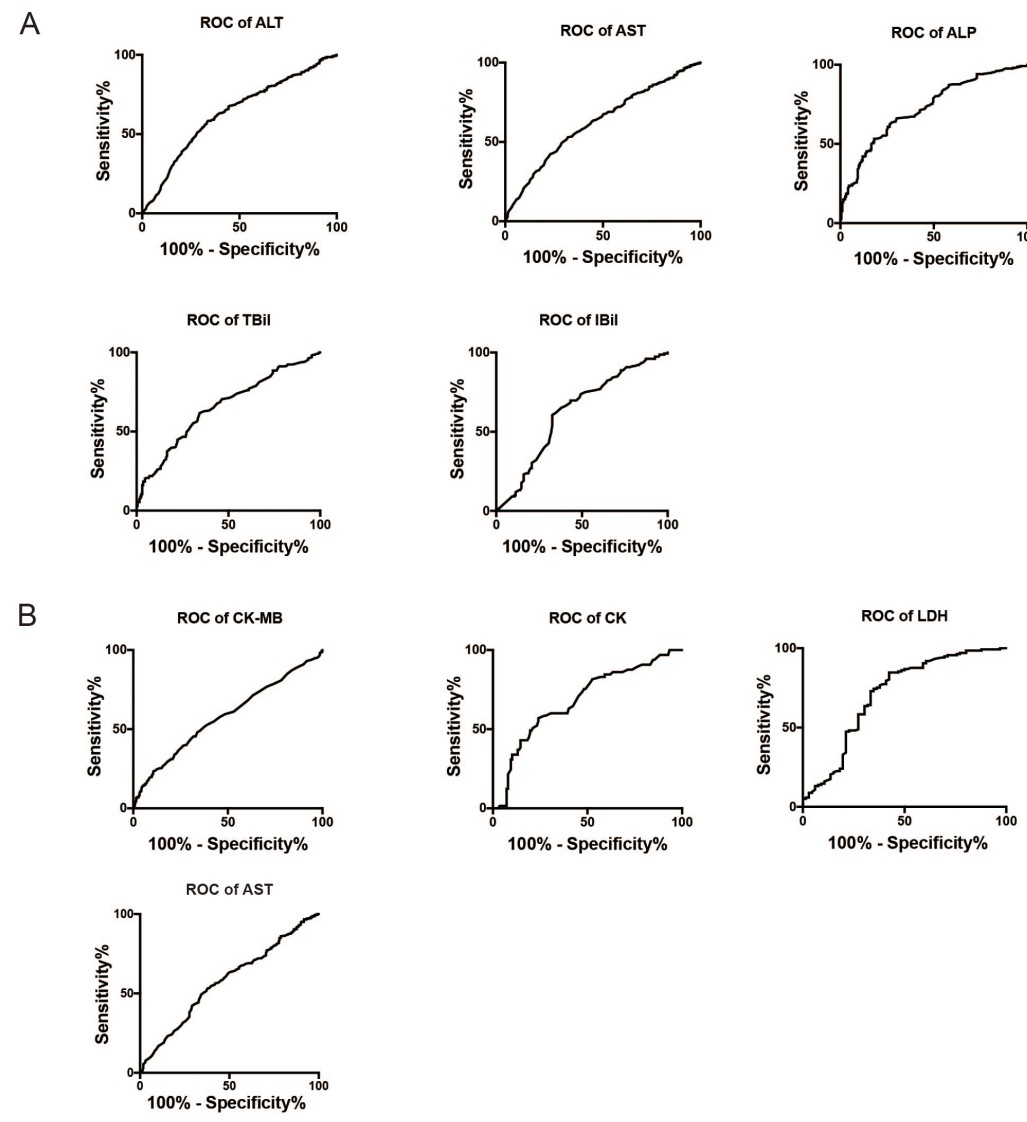

**Figure 3 ROC curves for biomarkers to assess liver or myocardial damage in infectious mononucleosis patients.** (A) ROC curves for ALT, AST, ALP, TBil, and IBil to assess liver damage in infectious mononucleosis patients. (B) ROC curves for CK, CK-MB, and LDH to assess myocardial damage in infectious mononucleosis patients. ALT, alanine transaminase; AST, aspartate transaminase; ALP, alkaline phosphatase; TBil, total bilirubin; IBil, indirect bilirubin; LDH, lactate dehydrogenase; CK, creatine kinase; CK-MB, creatine kinase-MB.

patients had active infection as confirmed by detection of the causative organism within 72 h of admission or by detection of EBV-IgM or IgM antibodies to other pathogens. All patients had features of the pediatric IM syndrome, and elevated levels of markers of active infection (*e.g.*, CRP). Patients with chronic infection were not included in our study cohort.

IM is a self-limiting disease, but it can lead to serious pulmonary, ophthalmic, nervous, hematological, hepatic, and myocardial damage (*Ishii et al., 2019*; *Greydanus & Merrick,*

**Table 4 Univariate and multivariate Cox proportional hazards regression analysis for liver damage.**

| Variables | Univariate | | | Multivariate | | |
|---|---|---|---|---|---|---|
| | HR | 95 CI | *P* | HR | 95 CI | *P* |
| Age | 1.10 | 0.79–1.53 | 0.512 | | | |
| >7 | Ref | | | | | |
| EBV infection | 1.14 | 0.79–1.72 | 0.446 | | | |
| EBV DNA+ | Ref | | | | | |
| ALT | 4.31 | 3.23–5.76 | <0.001 | 2.70 | 1.61–4.42 | 0.001 |
| >55.5 (U/L) | Ref | | | | | |
| AST | 5.26 | 3.95–7.01 | <0.001 | 3.23 | 2.12–6.54 | 0.001 |
| >56 (U/L) | Ref | | | | | |
| ALP | 2.31 | 1.77–3.02 | <0.001 | 2.19 | 1.08–3.44 | 0.003 |
| >206.5 (U/L) | Ref | | | | | |
| TBil (μmol/L) | 1.53 | 1.17–2.02 | <0.001 | 1.24 | 1.07–2.23 | 0.001 |
| >8.05 | Ref | | | | | |
| IBil (μmol/L) | 1.32 | 1.0–1.74 | 0.045 | | | |
| ≥3.65 | Ref | | | | | |

Notes.
HR, hazard ratio; CI, confidence interval; EBV, Epstein–Barr virus; ALT, alanine transaminase; AST, aspartate transaminase; TBil, total bilirubin; IBil, indirect bilirubin; ALP, alkaline phosphatase; LDH, lactate dehydrogenase; CK, creatine kinase; CK-MB, creatine kinase-MB.

**Table 5 Univariate and multivariate Cox proportional hazards regression analysis for myocardial damage.**

| Variables | Univariate | | | Multivariate | | |
|---|---|---|---|---|---|---|
| | HR | 95 CI | *P* | HR | 95 CI | *P* |
| Age | 1.91 | 1.26–2.88 | 0.003 | 1.86 | 1.34–3.05 | 0.001 |
| <3 | Ref | | | | | |
| EBV infection | 1.4 | 0.84–2.32 | 0.224 | | | |
| EBV DNA+ | Ref | | | | | |
| CK | 2.83 | 1.89–4.24 | <0.001 | 2.70 | 1.61–4.42 | <0.001 |
| >70.5 (U/L) | Ref | | | | | |
| CK-MB | 3.42 | 2.26–5.12 | <0.001 | 3.23 | 2.12–6.54 | <0.001 |
| >33.5 (U/L) | Ref | | | | | |
| LDH | 2.09 | 1.41–3.11 | <0.001 | 2.19 | 1.08–3.44 | 0.003 |
| >609 (U/L) | Ref | | | | | |
| AST>56.5 (U/L) | 0.91 | 0.60–1.37 | 0.632 | | | |

Notes.
HR, hazard ratio; CI, confidence interval; EBV, Epstein–Barr virus; LDH, lactate dehydrogenase; CK, creatine kinase; CK-MB, creatine kinase-MB; AST, aspartate transaminase.

*2019*; *Sako, Kenzaka & Kumabe, 2022*). Thus, the treatment focus is on managing symptoms and avoiding serious organ damage.

Although EBV-IM may occur at any age, the peak incidence is in the age-group of 2–4 years (*Kedi et al., 2019*). Similar to adult patients, most children recover without serious complications. Significant lymphocytosis, with high numbers of atypical lymphocytes, is a

## Hazard ratio of liver damage in IM patients

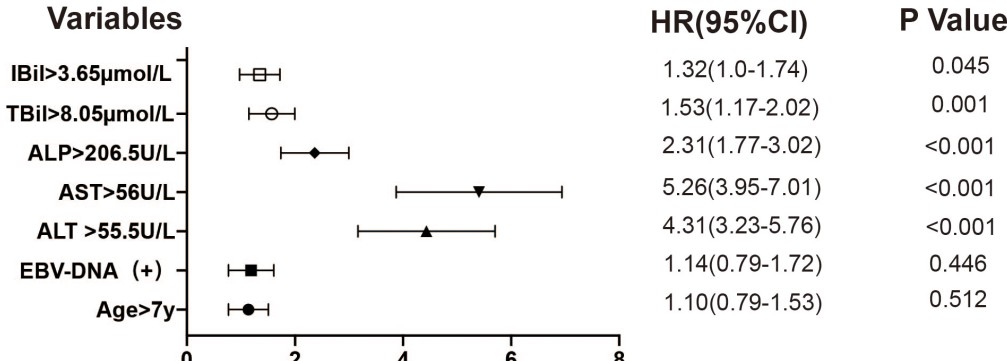

## Hazard ratio of myocardial damage in IM patients

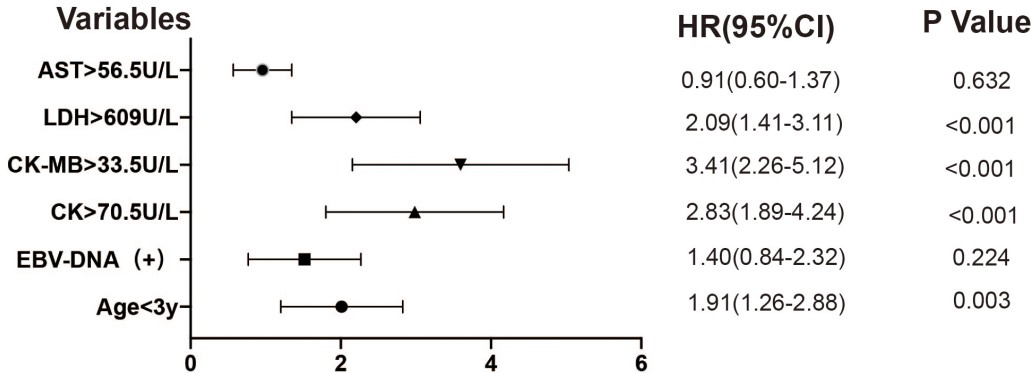

**Figure 4    Forest plots of age, EBV-DNA, and biomarker levels in infectious mononucleosis patients.**
The variables are to the left of the axis, while the *P* values are also shown on the right of the HR. EBV,
Epstein–Barr virus; CI, confidence interval; HR, hazard ratio.

consistent hematologic finding in all age-groups. The incidence of hepatitis is significantly correlated with age ($P = 0.003$) (*Sofie et al., 2015*). In the present study, age was also shown to be related to myocardial injury, with children below 3 years of age being more susceptible.

Because EBV is ubiquitous and lurks in lymphocytes, detection of EBV antibody alone is not sufficient to diagnose EBV-IM in young children. Identification of EBV-DNA in serum or plasma is the best test for diagnosis and monitoring of EBV infections and is also reported to be highly sensitive in young children (*Jiang et al., 2016*; *Holman et al., 2012*).

In our cohort, organisms causing IM included—in addition to EBV—CMV, *M. pneumoniae*, parvovirus, influenza virus, adenovirus infection, and chlamydia. We compared clinical characteristics between children with IM due to EBV *versus* children with IM due to other pathogens. EBV-DNA-positive IM was more frequent in the 7–17-years group than in the 0–3-years and 4–6-years groups, but EBV-IgM was comparable in the

different age-groups. EBV-DNA level in serum or plasma represents the EBV load and is related to disease progression. Older children may produce a stronger host immune response to EBV. In this study, EBV-DNA-positive IM patients had longer interval from symptom onset to hospital visit. Thus, the EBV DNA load before and after the hospital visit appear to be estimated of patients with EBV-IM. It is possible that the more severe symptoms (*e.g.*, painful pharyngitis and lymphadenopathy) in EBV-DNA-positive patients make them seek medical treatment sooner than EBV-DNA-negative patients. We also found that CMV is more likely to cause IM in infants and young children (0-3 years). Thus, for IM in young children, CMV infection is the primary consideration when EBV is negative.

EBV infects B cells, which activate cytotoxic T cells. This may lead to histopathological changes in several organs. Atypical lymphocytes, also known as Downey cells, are actually activated CD8 T lymphocytes, most of which are responding to EBV-infected cells (*Tang et al., 2014*). In this study, atypical lymphocyte counts were significantly higher in EBV-DNA-positive patients than in patients with IM due to other causes. CD3+ and CD8+ T lymphocytes counts were also significantly higher in EBV-DNA-positive patients, but CD4+ T lymphocytes counts were lower.

EBV-DNA detection by fluorescence quantitative polymerase chain reaction is a reliable tool for early diagnosis of primary EBV infection (*Yu et al., 2021*). EBV-DNA positivity represents active virus replication and can cause abnormal lymphocytosis. Among children with EBV-IM antibodies, we found that older patients had higher atypical lymphocytes counts and higher incidence of hepatitis; however, the younger patients (0–3 years) had higher risk for myocarditis. Thus, physicians should be concerned when there are changes in myocardial markers in children below 3 years with EBV-IM. Meanwhile, for IM patients over 7 years of age, changes in hepatic markers should alert the physician to possibility of liver damage, especially when EBV-DNA is positive.

Our study participants who were serologically confirmed to have EBV-IM or CMV-IM also underwent serological tests for detection of other causative organisms such as *T gondii*, HHV-6, HHV-7, adenovirus, rubella virus, herpes simplex virus, influenza/parainfluenza, rhinovirus, and coronavirus. Co-infection was a common phenomenon in patients with EBV-IM (over 50% of patients in this study). However, the present study only included those with serologically confirmed IM, so our results should be cautiously applied to patients with clinically diagnosed IM.

The severity of IM in children varies, so it is important to find effective evaluation indicators. At the early stage of the disease, clinical analysis of different blood biomarkers such as platelet, WBC, lymphocyte counts, can help determine whether there is bacterial or viral infection. For children infected with EBV, dynamic monitoring of ALT, AST, LDH, ALP, CK, and CK-MB levels can be useful for early identification of liver or myocardial injury. IM is a self-limiting disease with good prognosis and, generally, can be managed with only rest, good hydration and nourishment, and other supportive treatment. In children with serious symptoms, such as severe pharyngeal or laryngeal edema, nervous system complications, or myocarditis, a short course of use corticosteroids can reduce symptom severity. Patients with liver damage should be advised strict bed rest and be

managed according to the protocol used for viral hepatitis (*i.e.,* with hepatoprotective measures). In the early stages of IM, aggressive symptomatic and supportive treatment, especially for liver damage, can prevent its occurrence. Timely immunosuppressive and immunoregulatory therapy can significantly reduce the mortality rate. High fever or cervical lymph node enlargement, with increased WBC count with predominance of lymphocytes or monocytes or high proportion of atypical lymphocytes, increased liver enzymes or hepatosplenomegaly at the early stage of infection, or rapid progression or complications, are indications for antiviral therapy (*e.g.*, with ganciclovir) (*Naughton et al., 2021*). Active EBV infection and replication should be considered if EBV IgM antibody and DNA are positive, and the possibility of chronic active EBV infection or EBV-associated hemophagocytic lymphohistiocytic hyperplasia (*Kien & Ganta, 2020*) and other malignant neoplasms should be kept in mind if EBV-DNA copy number continues to increase.

In conclusion, our retrospective study revealed that younger age, shorter interval from symptom onset to hospital visit, cervical lymphadenopathy, tonsillar exudate, hepatosplenomegaly, atypical lymphocytosis, and moderate elevations of biomarkers could be used to differentiate IM caused by EBV from IM due to other pathogens. Evaluation of biomarkers and pathogen detection may allow physicians to take preventive actions to avoid serious complications in children with IM.

### Funding
This work was supported by grants from the Science Foundation of Wuhan Health and Family Planning Commission (grant number WZ22Q33 to Yangcan Ming) and (grant number WZ20M01 to Wen Su). The China Scholarship Council (CSC) provided financial support during Jing Yu's study abroad for this project. The funders had no role in study design, data collection and analysis, decision to publish, or preparation of the manuscript.

### Grant Disclosures
The following grant information was disclosed by the authors:
The Science Foundation of Wuhan Health and Family Planning Commission: WZ22Q33, WZ20M01.
The China Scholarship Council (CSC) provided financial support during Jing Yu's study abroad for this project.

### Competing Interests
The authors declare there are no competing interests.

### Author Contributions
- Yangcan Ming conceived and designed the experiments, prepared figures and/or tables, and approved the final draft.
- Shengnan Cheng performed the experiments, authored or reviewed drafts of the article, and approved the final draft.

- Zhixin Chen performed the experiments, authored or reviewed drafts of the article, and approved the final draft.
- Wen Su conceived and designed the experiments, authored or reviewed drafts of the article, and approved the final draft.
- Shuangyan Lu conceived and designed the experiments, prepared figures and/or tables, and approved the final draft.
- Na Wang analyzed the data, prepared figures and/or tables, and approved the final draft.
- Huifu Xu performed the experiments, analyzed the data, authored or reviewed drafts of the article, and approved the final draft.
- Lizhe Zhang performed the experiments, authored or reviewed drafts of the article, and approved the final draft.
- Jing Yu conceived and designed the experiments, prepared figures and/or tables, authored or reviewed drafts of the article, and approved the final draft.
- Jianqiao Tang conceived and designed the experiments, prepared figures and/or tables, and approved the final draft.

### Ethics

The following information was supplied relating to ethical approvals (i.e., approving body and any reference numbers):

This study was approved by the ethical committee of Wuhan Children's Hospital (Ethical approval number: 2021R034-E01), with waiver of the need for informed consent.

### Data Availability

The raw data are available as a Supplemental File.

### Supplemental Information

Supplemental information for this article can be found online at http://dx.doi.org/10.7717/peerj.15071#supplemental-information.

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
