# Peer review of "Infectious mononucleosis in children and differences in biomarker levels and other features between disease caused by Epstein–Barr virus and other pathogens: a single-center retrospective study in China"

_PeerJ, doi:10.7717/peerj.15071_

## Round 0.1 · original submission · Major Revisions

Both reviewers agree that your paper has some advantage for the publication of this journal. However, before that, several critiques should be corrected according to their advice. Please read carefully the comment and revise it based on one by one method.

Reviewer 1 ·

Basic reporting

no comment

Experimental design

no comment

Validity of the findings

no comment

Additional comments

#1 It is a wonderful study that mentions complications of IM in the evaluation of biomarkers. In children, there is a wide range of normal values for blood tests depending on age. What strategies did 0-3 and 7-17 year-olds have when comparing blood test results?

#2 On page 15, line 216, The mechanism of EBV suggests immune pathogenesis. Symptoms often appear with a delay, but when do you think abnormal blood test values occur and when do you think treatment should be considered?

·

Basic reporting

The structure of the paper is very good and professional with good English. There is a significant amount of biomaker data presented However, there are a number of issues which require addressing before the paper can be published.
All the biomarker parameters have been analysed in association with what the authors claim are the confirmed infectious agents or co-infections. Therefore the authors have to be accurate regarding the pathogen they claim infects the patients
1. The serological data presented for EBV, CMV and parvovirus B19 in minimal and the authors claim that 35.9% of patients were acutely co-infected with EBV and CMV. However that and due to the level of cross reacting antibodies or non-specific IgM generated during acute EBV infection, more information is required (ratios of sample to cut off for all assays) to confirm co-infection
2 Of the 1480 enrolled children 84.66 % had EBV, 10.74 % had CMV and 53.78% had acute mycoplasma infection. This indicates a significant level of co-infection and more detailed laboratory data needs to be presented to confirm this
3 I would be very suspicious of an EBV DNA detected result without any additional serological markers
4. Are the authors linking the biochemical parameters with acute primary infection or chronic infection
5. Does the indirect IFA for mycoplasma, influenza, RSV, adenovirus and parainfluenza detect viral antigens in a respiratory sample or serum antibody ?

Experimental design

More laboratory data needs to be presented to confirm the infectious agent before it can be linked to the biochemical parameters

Validity of the findings

As above

Additional comments

As above

---

## Round 0.2 · accepted · Accept

The authors have addressed all of the reviewers' comments.

The more critical reviewer did not respond to an invitation to re-review but I have assessed the revision by myself, and I am happy with the current version.

This manuscript is ready for publication.